# Accuracy of a Model-Free Algorithm for Temporal InSAR Tropospheric Correction

**Howard Zebker**

Department of Geophysics, Stanford University, Stanford, CA 94305, USA; zebker@stanford.edu;
Tel.: +1-650-723-8067

**Abstract:** Atmospheric propagational phase variations are the dominant source of error for InSAR (interferometric synthetic aperture radar) time series analysis, generally exceeding uncertainties from poor signal to noise ratio or signal correlation. The spatial properties of these errors have been well studied, but, to date, their temporal dependence and correction have received much less attention. Here, we present an evaluation of the magnitude of tropospheric artifacts in derived time series after compensation using an algorithm that requires only the InSAR data. The level of artifact reduction equals or exceeds that from many weather model-based methods, while avoiding the need to globally access fine-scale atmosphere parameters at all times. Our method consists of identifying all points in an InSAR stack with consistently high correlation and computing, and then removing, a fit of the phase at each of these points with respect to elevation. A comparison with GPS truth yields a reduction of three, from a rms misfit of 5–6 to ~2 cm over time. This algorithm can be readily incorporated into InSAR processing flows without the need for outside information.

**Keywords:** InSAR; InSAR calibration; InSAR validation; atmosphere variations; troposphere variations

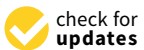

## 1. Introduction

Interferometric radar analyses follow from interpretation of the precise time delays and phase shifts in radar echoes as distance measurements, and then relating those geometric distances to topography or deformation of the surface. We generally assume that the signals propagate at a known constant velocity to convert the delays to distance. However, if the signals propagate through the Earth's spatially and temporally inhomogeneous atmosphere, which has a slightly higher index of refraction than free space, the velocity is lowered slightly, leading to variable delays in space and time which contaminate the observations.

In modern radar systems, atmospheric propagation variations are usually the largest artifacts present in interferograms, dominating SNR and decorrelation noises. Typical errors in a single interferogram from the variable water vapor part of the troposphere are ~2 cm rms [1], and can be several times greater across a time sequence as compared with the uncertainty in distance from phase estimation, i.e., at C-band, an SNR of 20 dB yields a phase or distance error of about 3 mm, even if a minimal one look is used. Hence, correcting for variations in the atmosphere is necessary in order to fully exploit the potential of spaceborne InSAR (interferometric synthetic aperture radar).

Many studies have addressed this problem specifically as regards InSAR analysis. Early work identified these phase artifacts [2–5], with [1,6–8] presenting the mapping from atmospheric parameters to InSAR phase. Ref. [1] also noted that the artifacts could most easily be compensated by stacking multiple observations, because models of the atmosphere at the time were not at a sufficiently fine resolution to correct the measurements. Quite a few papers have examined the use of both water vapor models (e.g., [9–13]) or empirically derived corrections from GPS and other instrumentation [13–18] to compensate InSAR images for propagation path delays. A comprehensive review is found in [13]. The conclusions from these works tend toward the following two main observations:

(i) the algorithms are often effective, but not universally so, and (ii) the reduction in total artifact energy is generally about a factor of two.

Many studies addressing compensation for propagation artifacts have focused on spatial variation and how to minimize it in individual interferograms. Temporal variations have been far less studied as regards InSAR analyses. Both [19] and [15] present sequences of GPS zenith wet delay that show variability with time, furthermore [15] and [20] compute temporal structure functions and power spectra of GPS zenith wet delay consistent with Kolmogorov-like $-5/3$ power law slopes. The spectra can grow quite large for long temporal intervals, consistent with decorrelation of the atmospheric delay over days-plus interferograms.

Some of the variations are well known to be readily compensated. The abovementioned and other works [21–23] generally show simple (approximately linear) relationships of atmospheric delay with elevation, hence, proposing that subtraction of that dependence "flattens" deformation interferograms with respect to topography. [24] further proposed using that correction routinely in the production of interferograms, in their case using the atmospheric term resulting from persistent scatterer analysis. Here, we validate and extend these efforts by assessing a set of Sentinel-1 InSAR data to verify and quantify the accuracy of temporal deformation observations using an algorithm that averages the elevation-residual phases in every interferogram and compensates for the long-time component of atmospheric variation. We show that a factor of three temporal artifact reduction can be obtained by an algorithm relating phase to elevation using only the InSAR data, so that the problems of assembling global high-resolution water vapor models can be avoided. More is yet to be gained in future work, likely by combining our proposed approach with some of the model-based solutions, and by ensuring multiple data acquisitions to average away tropospheric variations.

In the experimental results presented here, we use data acquired over the island of Hawaii during 2018. A large eruptive event occurred at Kilauea caldera in May 2018, leading to a decaying deformation signal over the following three months, and our work analyzes this deformation signal to illustrate our new method. The signal has decreasing effect as magma is propagated down the east rift zone, permitting comparison of the algorithm performance for a variety of signal magnitudes.

## 2. Materials and Methods

Denote the complex amplitude of a unit intensity plane wave at position $x$ in a medium with variable refractive index $n(x)$ and wavelength $\lambda$ by:

$$\mathrm{E} = e^{j(kn(x)x - \omega t)} \tag{1}$$

where the wavenumber $k = 2\pi/\lambda$. The accumulated phase along the propagation path is nearly

$$\phi = \int \frac{2\pi n(x)}{\lambda} dx. \tag{2}$$

For a radio signal propagating through free space, $n(x) = 1$, and the phase shift is directly proportional to path length as:

$$\phi = \frac{2\pi}{\lambda} x \tag{3}$$

The phase shift $\phi$ depends only on the free-space wavelength $\lambda$ and the distance propagated $x$. If the signal propagates instead through an atmosphere, $n(x)$ is not constant, and an additional phase shift follows. For the Earth's neutral atmosphere, $n(x)$ is always real and slightly greater than one, so we can expand $n(x)$ as $1 + 10^{-6} N(x)$, where $N(x)$, called the refractivity, is the additional refractive index due to the atmosphere. The change

in $n(x)$, from the free space value of one, is very small for Earth's atmosphere, hence, the factor of $10^{-6}$ in the definition. In this case, Equation (3) becomes:

$$\phi = \frac{2\pi}{\lambda}x + \frac{2\pi 10^{-6}}{\lambda}\int N(x)dx \qquad (4)$$

or

$$\phi = \frac{2\pi}{\lambda}x + \frac{2\pi}{\lambda}\Delta x \qquad (5)$$

where $\phi$ represents additional phase shift as a change in effective path length $\Delta x$. This value is often broken into two parts [25] as follows:

$$\Delta x = (\Delta x)_{hyd} + (\Delta x)_{wet} \qquad (6)$$

where $(\Delta x)_{hyd}$ and $(\Delta x)_{wet}$ represent the contributions to path length from the "hydrostatic" atmosphere and from water vapor. Empirical measurement of these effects [26] shows that Equation (6) may be approximated by:

$$\Delta x = 7.76 \times 10^{-5}\int_0^X \frac{P}{T}dx + 3.73 \times 10^{-1}\int_0^X \frac{e}{T^2}dx \qquad (7)$$

where $X$ is the total path length through the atmosphere, $P$ is the atmospheric pressure in millibars, $T$ is the temperature in Kelvins, and $e$ is the partial pressure of water vapor in millibars, and all of these quantities may vary with location along the propagation path. The constants preceding each integral are generally valid to within about 0.5% for frequencies up to 30 GHz and normal variations in pressure, temperature, and humidity.

Previous work by [1] noted that $P$ and $T$ change fairly slowly with location, while the inhomogeneity of water vapor ($e$) remains significant at small scales [8]. Furthermore, the path length, $X$, through the atmosphere depends on topography as propagation to sea level traverses more atmosphere than that to the top of a mountain.

Interferograms for temporal deformation studies are acquired at different points in time, and thus any temporal phase (delay) difference creates tropospheric artifacts. While pressure or temperature may vary slowly with location, because InSAR component images are acquired days apart, these quantities can be quite different, and this difference will depend on elevation. The wet troposphere delay will vary both spatially and temporally, with the longer spatial scale components behaving like $P$ and $T$ and the short spatial scale artifacts persisting as highly variable and obvious noises. In addition, in the presence of severe weather, these quantities can vary significantly over time scales as short as a few hours, and the general trend of decreasing delay with elevation may even be reversed due to local atmospheric conditions.

None of the present water vapor models are very accurate at m-scale resolution, hence, only averaging can reduce these. Fortunately, most of the signal power in the phase distortions is at long scales due to their power law expression. The atmosphere exhibits a rotationally averaged spatial power spectrum of phase delay that decreases approximately as a power law [6,8] with slope $-5/3$ at length scales from an outer scale the size of the radar scene (~100 km) to the wet troposphere scale height of 1–2 km [6], and with slope $-8/3$ down to an inner scale of the SAR resolution (~5 m). Integrating the power spectrum of the artifacts using these values shows that 97% of the energy is at the long scales with only 3% at the short scales, hence, the value of the correction procedures. Next, we discuss the goal of most tropospheric correction algorithms, which is to remove, as much as is feasible, the longer scale artifacts, with the smaller scale distortions reduced via averaging [5].

## 3. Results

### 3.1. Correction Algorithm

Before presenting our correction algorithm, it is useful to note the steps needed for interrelating a stack of radar interferograms. The processing flow we use here is the following:

1. Geocode and phase compensate single-look complex (SLC) images;
2. Form interferogram pairs as desired and unwrap;
3. Choose reference point(s) to remove artifacts due to unwrapping zero phase point;
4. Remove troposphere using height regression;
5. Compute time series using the SBAS method (small baseline subset analysis [27]).

Steps 3 and 4 define our atmospheric correction algorithm. Step 3 is needed for time series analysis because each interferogram may contain an arbitrary phase constant. Phase unwrapping consists of computing phase gradients, and then integrating the result, thus, all derived InSAR phases are relative to the choice of the reference point. In addition, if the reference point itself is moving all inferred time series will reflect this motion. Step 4 involves correcting the set of interferograms for elevation-dependent temporal variations in pressure, temperature, and the long spatial scale component of water vapor.

Step 5, the temporal dependence estimator, can incorporate the final atmosphere-reducing procedure, namely averaging multiple scenes to lessen the variable water vapor artifacts at all scales.

### 3.2. Reference Point Selection

If we are to analyze successive temporal measurements of InSAR phase, we must separate motions of the surface from the propagational phases due to the variable atmosphere. Again, unwrapping the interferogram phases in each image loses the constant term, because each interferogram consists only of relative values. In most cases, a location assumed to be non-moving is chosen because the entire derived interferogram would appear to move in unison if the reference point itself is not stationary.

Even if the selected reference point does not move, or has known motion, the reference will undergo apparent motion because it too is imaged through the variable atmosphere. Thus, there are two challenges to selecting the reference, i.e., we need outside knowledge of the scene to know where the nonmoving points may be, and the reference exhibits apparent motion from phase artifacts. The latter effect may be minimized if we select multiple reference points and average the results. This compensates for the short length-scale variations, and results in an average phase offset for the image. By subtracting the average offset from its interferogram in the next step, the long wavelength portion is removed as well.

Our approach is to automatically select reference points by finding all of the points in a scene that are reliably estimated, specifically, those with high correlation values. We identify these by examining the correlation at each point in all interferograms forming a stack, and we choose only those points whose correlation exceeds a threshold in every interferogram. It is important that the references in every interferogram have reliably estimated phase, or else the scene will add arbitrary temporal noise to the stack. Because terrains differ, the appropriate threshold will vary in different stacks. For better correlated scenes, a value of 50–60% is adequate, while for more highly vegetated scenes a value as low as 20–30% may be needed. We typically examine thresholds that identify several hundred to several thousand reference points, ensuring a wide enough spatial distribution to average out spatial artifacts.

We illustrate, in Figure 1, reference point selection in a set of Sentinel-1 data acquired over the island of Hawaii. The background image is the average correlation, which ranges from near zero in the water and in the densely forested areas, to nearly unity on the bare lava surfaces. The green squares are the chosen reference points, which are located primarily in the high correlation regions and are well distributed spatially.

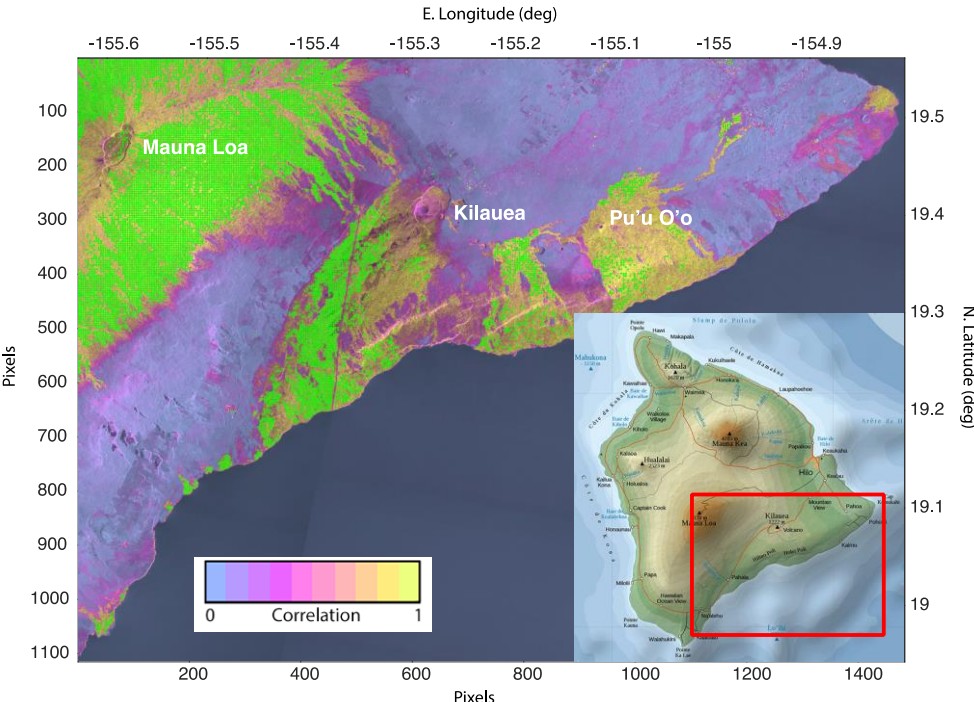

**Figure 1.** Reference locations from a stack of interferograms over Hawaii. The data comprise 50 single-look complex (SLC) scenes acquired from January to December 2018, and the stack in this image is formed from all possible interferograms with time baselines of 12 days or less. The green squares, which are not part of the colorbar for correlation, show the reference points, which are the points whose correlation exceeds 50% in all interferograms. Note that portions of Hawaii are highly vegetated, and lead to the large regions of consistent decorrelation (purple color) in the image.

### 3.3. Troposphere Height Regression

Once we have selected the set of reference points, we compute the phase correction function. A geographically distributed set of reference points averages spatial variations of delay, but the points lie at many different elevation levels, and the propagation path length through the atmosphere will vary. We noted that the correction for spatially diverse variations in $P$, $T$, and $e$ will be dependent on elevation, so we compute an unweighted linear regression of the phase at each point against its topographic height rather than simply average all reference phases. In this instance, a simple linear fit suffices to remove the artifacts, although a higher order fit may be used if desired. We compute the elevation fit for each interferogram in the stack and subtract that function from the observed phases. In this way, each interferogram is referenced to a common zero level. This approach is valid as long as the image area undergoing significant deformation is small as compared with the full scene. Here, roughly 550 points are in the deforming region, out of 9486 total references. Hence, their influence on the result is small, but indeed remain as an error source.

The results of the correction can be seen in Figure 2, where we compare several interferograms from the Hawaii stack described above, before and after the elevation correction. The corrected interferograms (b) show much more consistent phases than the uncorrected values (a) as the large spatial scale phase change from temporal atmospheric variations is removed. This can be readily seen in the final time series result over a nondeforming area (see the statistics for GPS station UWEV in Table 1 below). The corrected interferograms form the input data set for an SBAS temporal solution.

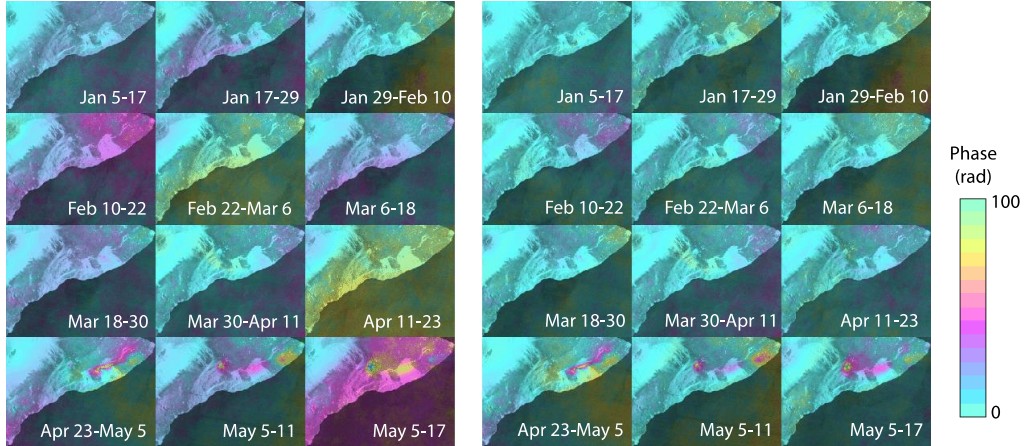

**Figure 2.** Sequence of 12 example interferograms (January–May 2018) from Hawaii stack, before (at left) and after (at right) regression correction. The corrected images display much less variation with time than the uncorrected images, hence, the derived time series will be more accurate as the average phase of each scene is equalized. The color scale repeats every 100 radians so that both large and small deformation are visible.

**Table 1.** Average RMS GPS/InSAR error, cm. For each site, rms difference is computed (as in Figure 4) both (with) using the atmosphere correction described here, and (without) using the standard single reference point algorithm. Results are shown for SBAS solutions with maximum temporal baselines of 12, 30, and 100 days. The 100-day solutions reduce the atmosphere more through averaging but lose some signal from undersampling the phase gradient (see text).

| RMS Errors (cm) | | 12 Day | | 30 Day | | 100 Day | |
|---|---|---|---|---|---|---|---|
| GPS Site | Height (m) | with | without | with | without | with | without |
| MOKP | 4133 | 1.7 | 8 | 1.8 | 6.9 | 1.5 | 6.3 |
| PAT3 | 3831 | 1.3 | 7 | 1.4 | 6.1 | 1.2 | 5.6 |
| MLES | 3841 | 1.3 | 7.7 | 1.4 | 6.6 | 1.2 | 5.9 |
| OUTL | 1105 | 1.8 | 5.1 | 4 | 3.5 | 7.2 | 5.4 |
| AHUP | 1105 | 2.7 | 8.1 | 1.5 | 5 | 3.4 | 2.5 |
| MLSP | 4078 | 1.7 | 7.9 | 1.7 | 6.9 | 1.5 | 6.2 |
| PUOC | 893 | 4.8 | 2.7 | 2.4 | 3.4 | 2.9 | 1.9 |
| KOSM | 990 | 1.4 | 5.7 | 1.3 | 4.6 | 1.4 | 3.6 |
| UWEV | 1257 | 1.6 | 5.2 | 1.4 | 3.9 | 1.4 | 3.0 |
| CNPK | 1124 | 2.9 | 4.3 | 2 | 3.6 | 1.6 | 2.9 |
| KAMO | 781 | 2.9 | 8 | 3.1 | 6.6 | 3.3 | 5.9 |
| Mean | | 2.2 | 6.3 | 2 | 5.2 | 2.4 | 4.5 |

### 3.4. Validation Using GPS as Reference

In this section, we quantify the correction of several time series from the 2018 Hawaii Sentinel dataset and compare with GPS receiver deformations. A map of the GPS locations on the InSAR cumulative displacement image for 2018 is shown in Figure 3. Numerical values for the rms errors at each site are given in Table 1.

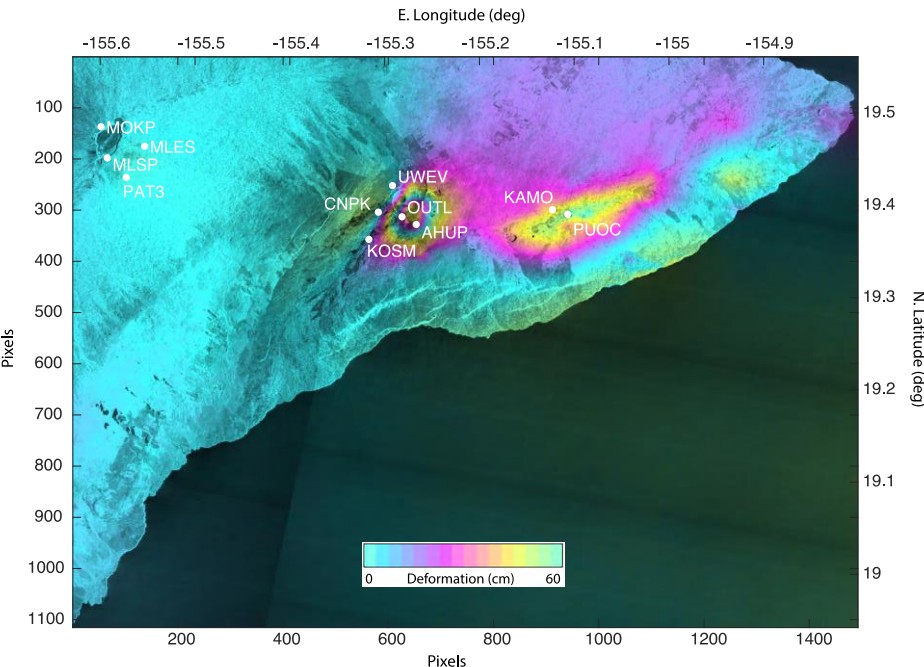

**Figure 3.** GPS locations on the InSAR cumulative displacement image for part of Hawaii during 2018. Areas covered are the summit of Mauna Loa, the Kilauea area, and the Pu'u O'o region. While other GPS sites are maintained, the areas are the most instrumented.

We chose a list of 11 GPS locations covering disparate regions on Hawaii, near the top of Mauna Loa, around Kilauea, and the Pu'u O'o area, in order to examine how the correction performs at different scene locations. In each case, we derived an SBAS time sequence using a set maximum temporal baseline, with and without the algorithm described here, and displayed the InSAR result along with the radar line of sight component of the GPS solution (Figure 4). All GPS data are plate-fixed 24 h solutions provided by the Nevada Geodetic Laboratory GPS repository [28].

In nearly all cases the corrected solution errors (denoted with) are smaller than the single reference approach. Furthermore, the improvement is greatest for the sites near the Mauna Loa summit, demonstrating the need for elevation dependence in the correction. Looking at the mean rms values for each processing stream, it is clear that the gain is greater for shorter maximum baselines. For a longer baseline limit, more interferograms span a given time, and their averaging in the SBAS reduction helps mitigate the atmospheric artifacts. The overall corrected error is not very sensitive to baseline parameters because the longer baselines underestimate deformation in large-motion areas (e.g., stations OUTL and PUOC, where the improvement is a factor of two at best) due to temporal aliasing of the raw interferometer fringes at the 60 m spatial sampling used here (personal communication, K. Pepin, Stanford University). This added smoothing compensates a bit for the atmospheric noise, albeit at loss of signal.

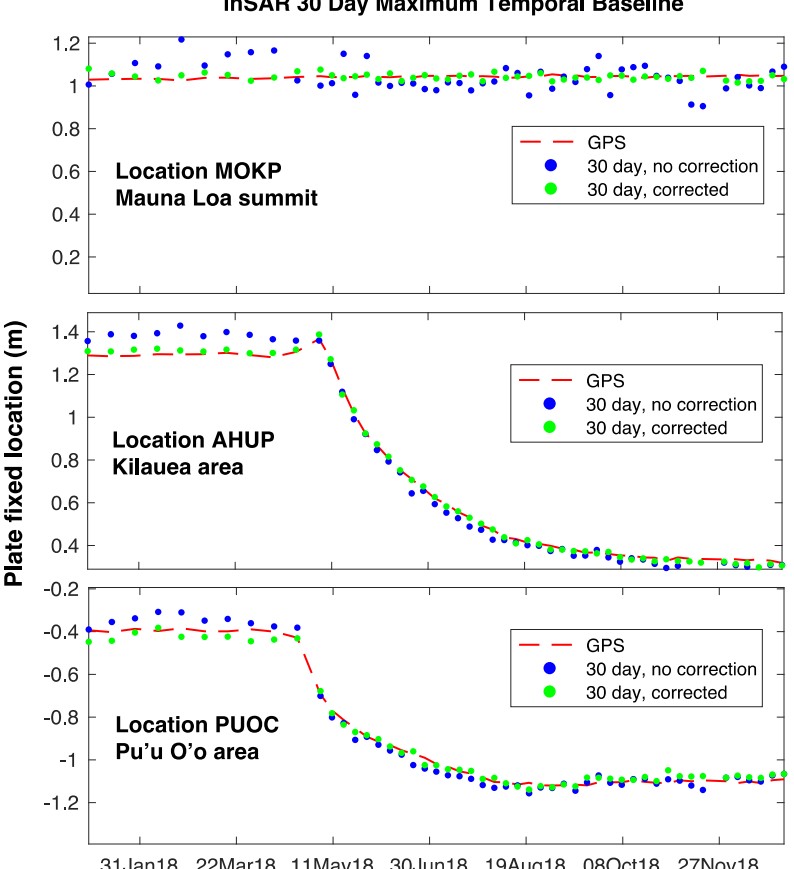

**Figure 4.** GPS/InSAR comparisons in the three study areas. Dashed red line is plate-fixed GPS location, blue dots are InSAR time series without the proposed atmosphere correction, and green dots are with the correction applied. Solutions here are for an unconstrained SBAS solution with maximum 30-day temporal separation, values for remaining GPS sites and with maximum baselines of 12 and 100 days are summarized in Table 1.

## 4. Discussion

The dominant source of error for InSAR time series imaging, phase artifacts from the variable atmosphere, can be compensated with an algorithm that requires only the InSAR data. A comparison with GPS measurements suggests that a reduction of a factor of three in rms misfit is often possible, which is equal to or better than many weather model-based methods, and avoids the need to globally access fine-scale atmosphere parameters at all times. Our method is automatic in the sense that it consists of identifying all points in an InSAR stack with consistently high correlation and computing a fit of the phase at each of these points in each interferogram with respect to elevation prior to temporal analysis. Thus, this algorithm can be readily incorporated into InSAR processing flows without the need for outside information. Future work promises to reduce these noises further if the advantages of local weather data can be reliably added as a final step.

**Funding:** This research was funded by NASA, grant numbers NNX17AE03G and NSSC19K1485.

**Institutional Review Board Statement:** Not applicable.

**Data Availability Statement:** All Sentinel data used here are available from the Alaska Satellite Facility as part of the NASA radar DAAC. The GPS data may be obtained from the Nevada Geodetic Laboratory at the University of Nevada, Reno, USA.

**Conflicts of Interest:** The author declares no conflict of interest. The funders had no role in the design of the study; in the collection, analyses, or interpretation of data; in the writing of the manuscript, or in the decision to publish the results.

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
