# Peer review of "Accuracy of a Model-Free Algorithm for Temporal InSAR Tropospheric Correction"

_remotesensing, doi:10.3390/rs13030409_

Round 1

Reviewer 1 Report

line 46-48: confusing text

line 123: unclear , please rephrase

line 150: define SBAS and give reference

line 212: replace "will be" with "is"

line 213: what is small ? what is a full scene ? give orders of magnitude and explain why

line 219: what do you mean with "more accurate" ?

line 223: give details and magnitudes about the "more consistent phases".

line 227: replace "truth" with "reference". GPS does not give the truth, only a measurement used as reference.

line 241: remove "highly"

line 288: replace "as sensitive" with "very sensitive" and provide values

line 297: replace "GPS truth" with "GPS measurements"

Improve english style. Avoid the use of "we" and "our"

Author Response

Reviewer 1.

line 46-48: confusing text

The identified text is now more exact, and less colloquial.

line 123: unclear , please rephrase

Reworded.

line 150: define SBAS and give reference

Reference added.

line 212: replace "will be" with "is"

Done.

line 213: what is small ? what is a full scene ? give orders of magnitude and explain why

More detail included.  We have computed how many points are in the deforming region (550) and how many total reference points were used (9486), hence their influence on the total result is small, but remains as an error source.

line 219: what do you mean with "more accurate" ?

Added text in caption to explain why removing large spatial scale phases helps.

line 223: give details and magnitudes about the "more consistent phases".

Added an example of the reduced variance of the phase in a non-deforming area, at GPS location UWEV.

line 227: replace "truth" with "reference". GPS does not give the truth, only a measurement used as reference.

Done.

line 241: remove "highly"

Agree, done.

line 288: replace "as sensitive" with "very sensitive" and provide values

Value given now.

line 297: replace "GPS truth" with "GPS measurements"

Done.

Improve english style. Avoid the use of "we" and "our"

This is a stylistic choice.  While we appreciate the reviewer’s point of view, we elect to use an active voice rather than a passive voice to discriminate exactly what we have done as opposed to what has been done by others.  We respectfully prefer to decline this suggestion, but if the editorial style of the journal insists we can change the voice.  We think it is better described as we have written it.

Reviewer 2 Report

The main remark is that the experimental area is not described. It is necessary to give a physical-geographical and climatic description. It is advisable to present a map with relief contours and areas of forest vegetation.

The May 2018 earthquake in Hawaii, which determines the shape of the graphs in Figure 4, is not even mentioned in the article.

Detail remarks:

  1. p. 2, line 96. Error in the formula 4, the second term must be an integral of N(x) over dx
  2. p. 2, line 105. The author confuses hydrostatic delay (i.e. proportional to the TOTAL atmospheric density and TOTAL pressure) and dry delay (i.e. proportional to the density of dry gases). Based on the formula (7), the speech in the article is about the hydrostatic delay
  3. p. 4, line 150. Method SBAS comes out of nowhere, no explanation is given in the text
  4. p. 5, line 203. The altitude correction algorithm is not entirely clear. Could the author give formulas?
  5. p. 6, fig 3. It's necessary to turn the figure.
  6. p. 7, fig 4 and table 1. Obviously, It's necessary to indicate not "GPS location", but "GPS site height"

Author Response

Reviewer 2:

The main remark is that the experimental area is not described. It is necessary to give a physical-geographical and climatic description. It is advisable to present a map with relief contours and areas of forest vegetation.

The inset map in Fig. 1 shows the elevation range, and we have increased its size a bit to make it easier to read.  We also modified the caption to point out that vegetation density follows from the decorrelated areas in the image so that readers can see where this is a problem.

The May 2018 earthquake in Hawaii, which determines the shape of the graphs in Figure 4, is not even mentioned in the article.

We have added a description of the geologic context to the introduction emphasizing the reviewer’s point.

Detailed remarks:

  1. 2, line 96. Error in the formula 4, the second term must be an integral of N(x) over dx

Yes, this was a simplification and we have rewritten it as suggested.

  1. 2, line 105. The author confuses hydrostatic delay (i.e. proportional to the TOTAL atmospheric density and TOTAL pressure) and dry delay (i.e. proportional to the density of dry gases). Based on the formula (7), the speech in the article is about the hydrostatic delay

If I understand the question, the formulas as given reflect both the wet and dry troposphere, both of which can change over time and space.  Our method addresses total delay, wet or dry, so it is the sum that must be considered.  This dependence should be clear from eq. 7, and the text does note that P and T behave spatially rather differently than e.  If this is still unclear please point out the confusing text and will try to clarify it.

  1. 4, line 150. Method SBAS comes out of nowhere, no explanation is given in the text

reference now given explicitly.

  1. 5, line 203. The altitude correction algorithm is not entirely clear. Could the author give formulas?

The text has been extended to better describe the method in detail.  As it’s a simple linear fit, we don’t think an equation is needed, but we can add one if the new text is not sufficiently clear.

  1. 6, fig 3. It's necessary to turn the figure.

Yes, our blunder. Fixed.

  1. 7, fig 4 and table 1. Obviously, It's necessary to indicate not "GPS location", but "GPS site height"

Heights added to table for use in assessing elevation range.

Reviewer 3 Report

Review of the “Accuracy of a Model-Free Algorithm for Temporal InSAR Tropospheric Correction” Technical Note. In this letter, an algorithm is described to correct the tropospheric signal for temporal InSAR product using only the InSAR data. I believe that this letter can be published after fixing minor things, mainly the figures.

This letter it’s well written, the methodology is well described, but there are some weaknesses.

  • There is a lack of references, for example, line 29 or line 33 (~2 cm); line 43 can also be improved. A reference for the refractivity constants used in equation 7 is necessary.
  • Line 124-125, it is not valid in the case of severe weather. Pressure and temperature can, in the same location, derive a lot in a few hours.
  • Line 126-127, we tend to consider this statement true. However, I find quite a few cases where the wet quantity is higher in the upper layers and lower in the lower layers. Clearly, it depends on the region (characteristics of the mountains), the atmospheric conditions, among other causes. But this fact must be taken into account in the text.
  • Figure 1: Where is the green in the colorbar?
  • Figure 2: 0 and 100 rad have the same color?
  • Figure 3: is it upside down?
  • Figure 4: Explain the decay of the curve after 11May18.
  • A comparison is lacking between the results obtained with this method and what would be obtained using external information, from a weather forecast model, for example. This would be the ultimate way to show that the technique works and has better results before speaking about incorporating this technique in the InSAR processing flows method.       

Author Response

Reviewer 3.

There is a lack of references, for example, line 29 or line 33 (~2 cm); line 43 can also be improved. A reference for the refractivity constants used in equation 7 is necessary.

Thank you for showing where assumptions have been introduced without support. Reference for 2 cm included, and the values in equation are in the reference to Smith and Weintraub as given.  There are no numbers in line 29, only a statement of general physics, so I do not see what ought to be referenced for this point.  Any general e&m physics text would have this statement, so it should not require a specific reference.  If the referee insists we can point to a physics textbook.

Line 124-125, it is not valid in the case of severe weather. Pressure and temperature can, in the same location, derive a lot in a few hours.

Yes, this is true, and we have edited the text to specifically state this.

Line 126-127, we tend to consider this statement true. However, I find quite a few cases where the wet quantity is higher in the upper layers and lower in the lower layers. Clearly, it depends on the region (characteristics of the mountains), the atmospheric conditions, among other causes. But this fact must be taken into account in the text.

We agree. Again, we have edited the text to make this point more clearly.

Figure 1: Where is the green in the colorbar?

There is no green in the colorbar, that is why this color was used.   The definition of the green points is in the figure caption, and mentioned specifically.

Figure 2: 0 and 100 rad have the same color?

Yes, this can be confusing, we agree.  The deformation is shown on a cyclic colorbar to accommodate large changes as well as smaller ones with a single scale.  This is common for phase plots, and we added an explanation to the figure caption to make this more clear.

Figure 3: is it upside down?

Fixed.

Figure 4: Explain the decay of the curve after 11May18.

This deformation pattern is expected and explained now in the introduction where the geologic context is added.

A comparison is lacking between the results obtained with this method and what would be obtained using external information, from a weather forecast model, for example. This would be the ultimate way to show that the technique works and has better results before speaking about incorporating this technique in the InSAR processing flows method.       

In this paper we have described an approach to correcting for tropospheric variations without requiring a weather model.  We include many references to experiments that indeed show the success of using a weather model.  It is beyond the scope of the current work to duplicate the results of the various weather model studies.  If the referee is unsatisfied with including these results by reference, we can summarize these many studies, but that would make this paper a review of others’ work rather than presenting an alternate approach.  We would prefer to concentrate on our work here.

Round 2

Reviewer 2 Report

Dear author,

Replace (∆x) dry in formula 6 to (∆x) hyd.

Replace the phrase "where (∆x) dry and (∆x) wet represent the contributions to path length from the" dry "atmosphere and from water vapor" to " where (∆x) hyd and (∆x) wet represent the contributions to path length from the hydrostatic atmosphere and from water vapor".

Remarks:

In Formula 7, the first term denotes hydrostatic hold, not dry hold. To understand this, you can consult the numerous specialized literature on GPS meteorology.

The simple reason is that P is the TOTAL pressure of atmospheric gases, including water vapor, and not the pressure of ONLY dry gases (nitrogen and xyrogen)

Author Response

Response to Reviewer 2 second round comment:

Thank you for this comment.  We have changed the terminology from “dry” to “hydrostatic” as suggested.  This makes our notation and terminology more consistent with the bulk of the GPS literature.  Our original notation derived from 1950’s study of the atmosphere, and this change brings it up to contemporary standards.  All changes included as suggested.